# A Novel Nomogram Based on Initial Features to Predict BPH Progression

**DOI:** 10.3390/ijerph19159738

**Published:** 2022-08-08

**Authors:** Lorenzo G. Luciani, Daniele Mattevi, Daniele Ravanelli, Umberto Anceschi, Guido Giusti, Tommaso Cai, Umberto Rozzanigo

**Affiliations:** 1Robotic Surgery Unit, S. Chiara Hospital, 38122 Trento, Italy; 2Department of Urology, S. Chiara Hospital, 38122 Trento, Italy; 3Health Physics Unit, S. Chiara Hospital, 38122 Trento, Italy; 4Department of Urology, IRCCS National Cancer Institute, Regina Elena, 00144 Rome, Italy; 5Department of Urology, IRCCS S. Raffaele Hospital, 20132 Milan, Italy; 6Department of Radiology, S. Chiara Hospital, 38122 Trento, Italy

**Keywords:** benign prostatic hyperplasia, BPH, lower urinary tract symptoms, LUTS, progression, alpha-blockers, 5-alphareductase inhibitors, 5-ARI, phytotherapy, supplements, *Serenoa repens*

## Abstract

Objectives: The aim of this study was to establish a tool to identify patients at risk for pharmaceutical and surgical interventions for benign prostatic hyperplasia (BPH)-related lower urinary tract symptoms (LUTS) over a 10 year follow-up. Methods: The data of patients with mild to moderate male LUTS undergoing phytotherapy from January to December 2010 were reviewed. Patients were followed for 10 years through medical visits and telephone consultations. The outcomes were (1) treatment switch from phytotherapy or no therapy to alpha-blockers or 5α-reductase inhibitors (5-ARI), and (2) clinical progression (acute urinary retention or need for surgery). Two calibrated nomograms (one for each outcome) were constructed on significant predictors at multivariate analysis. Results: A total of 107 patients with a median age of 55 years at presentation were included; 47% stopped or continued phytotherapy, while 53% switched to alpha-blockers and/or 5-ARI after a median time of 24 months. One-third in the second group experienced clinical progression after a median time of 54 months. Age, symptom score, peak flow rate (Qmax), prostate-specific antigen (PSA), and post-void residual volume were significantly associated with the outcomes. According to our nomograms, patients switching therapy or progressing clinically had average scores of 75% and 40% in the dedicated nomograms, respectively, as compared to 25% and <5% in patients who did not reach any outcome. Conclusions: We developed a nomogram to predict the risk of pharmaceutical or surgical interventions for BPH-related LUTS at 10 years from presentation. On the basis of our models, thresholds of >75% and >40% for high risk and <25% and <5% for low risk of pharmaceutical or surgical interventions, respectively, can be proposed.

## 1. Introduction

Benign prostatic hyperplasia (BPH) is a progressive condition that becomes a clinical entity if and when it is associated with subjective symptoms, the most common manifestation being lower urinary tract symptoms (LUTS) or urologic complications [1]. The cumulative incidence of acute urinary retention (AUR) and the need for BPH-related surgery were 2.7% and 3% over 4–6 years in the Olmsted county survey, a cohort of 2115 men aged 40–79 years [2]; their rate increased with age. The progression from male lower urinary tract symptoms (LUTS) to clinically relevant benign prostatic hyperplasia (BPH) is difficult to predict. Multiple risk factors for BPH progression have been identified; prostate volume, prostate-specific antigen (PSA), peak flow rate (Qmax), post-void residual volume, age, and symptom scores have been associated with an increased risk of clinical progression [3]. The variety of clinical presentation, the imperfect overlapping between LUTS and BPH, and the presence of other medical conditions misleading or delaying the diagnosis can make it difficult to predict which patients will progress and when. Phytotherapy is available in a number of herbal drug preparations, and it is often chosen as a first-line treatment for mild to moderate LUTS [4,5]. However, how and when patients switch to pharmaceutical/surgical interventions, and which conditions predispose to such events over a long-term period have not been fully elucidated. Identifying patients at highest risk for progression can improve the decision making in each individual and the selection of those who would benefit from such interventions. A nomogram is a model that uses an algorithm or mathematical formula to predict the probability of an outcome, maximizing the predictive power of variables through the convergent use of all important data parameters. Nomograms incorporating diagnostic information can provide personalized, evidence-based answers to key questions. Our objective is to establish a tool to identify patients at risk for pharmaceutical and surgical interventions on a 10 year follow-up.

## 2. Materials and Methods

### 2.1. Study Design

The data of male patients presenting with LUTS from January to December 2010 were retrospectively reviewed. All patients were assessed at presentation and given phytotherapy as a first-line approach.

### 2.2. Definition and Inclusion/Exclusion Criteria

The categorization of LUTS into storage, voiding, and post-micturition by the International Continence Society (ICS) was used [6]. Only patients naïve for any urologic treatment were considered; patients with a previous single episode of urinary tract infection (UTI) were also included for analysis. On the other hand, patients with previous documented recurrent UTI, previous urologic surgery, severe symptoms, and developing prostate cancer during the follow-up years were excluded.

### 2.3. Patients’ Workout and Treatment

Patients’ workout comprised a baseline physical examination including a rectal examination, urine culture, uroflowmetry, prostate volume and post-void residual volume at abdominal ultrasound, PSA, and International Prostate Symptom Score (IPSS). The phytotherapy regimens included 3 month cycles of phytotherapy, based on the extract of *Serenoa repens* (hexanic or ethanolic), alone or in combination with other plant extracts, i.e., cranberry, *Boswellia*, *Curcuma*, *Urtica*, or 1 month cycles of pollen extract. The medical treatment for BPH-associated symptoms includes alpha-blockers, 5α-reductase inhibitors, and anticholinergics.

### 2.4. Definition of Outcomes

Typically, three scenarios are expected: (a) symptoms stop: phytotherapy is discontinued or repeated; (b) symptoms persist or progress to moderate: switch to synthetic drugs; (c) clinical progression: catheter placement for AUR or need for surgery. The first outcome investigated was treatment switch from phytotherapy to alpha-blocker or 5-ARI (outcome 1: step from (a) to (b)). The second outcome measured was clinical progression (outcome 2: step from (a) or (b) to (c)). Symptoms were assessed by IPSS. Clinical progression was defined as acute urinary retention with catheter placement, or the need for surgery, based on severe symptoms and/or increasing residual volume to >150 mL. Patients were followed at a 6 month interval over a 10 year span. Patients with clinical progression were not further considered in the study. Follow-up was rescheduled or re-modulated during the COVID-19 outbreak through telephone or email consultations, as previously described [7].

### 2.5. Statistical Analysis

Continuous variables are reported as the median with SD; categorical variables are expressed as absolute numbers (*n*) and proportions (%); the Kruskal–Wallis test was performed to compare the variables in the patients’ subclasses according to the outcomes. To develop a well-calibrated nomogram to predict the outcomes, we performed univariate (UVA) and multivariate (MVA) logistic regression analyses to screen for predictors of clinical progression (age, IPSS, Qmax, prostate volume, PSA, and post-void residual volume). UVA was performed to explore all the variables and select a dataset of potential predictors (*p* < 0.05). Subsequently, MVA was used to establish the nomogram of the prediction model. The nomogram was validated internally using a calibration curve, whereas the area under the curve (AUC) of receiver operating characteristic (ROC) analysis was used to evaluate the discriminative performance of the nomogram. The nomogram is used by locating a patient’s position for each variable on the horizontal scale. A point value is assigned according to the point scale (top axis) and summed for all variables. The total points correspond to a probability value for experiencing treatment switch or clinical progression. All statistical analyses were carried out on R V.3.6.2 (R Development Core Team (Vienna, Austria)). The packages of rms and pROC were involved in this process.

## 3. Results

### 3.1. Case Series

A total of 107 male patients with a median age of 55 years referred for male LUTS completed at least one cycle of phytotherapy in the above period. Table 1 shows the patients’ characteristics. The main complaints were storage (*n* = 57) and voiding symptoms (*n* = 26), also combined (*n* = 23); post-micturition (*n* = 3) and LUTS associated with pelvic/perineal pain (*n* = 15) were also reported. As far as outcomes are considered, 50 (46.7%) of 107 patients stopped or continued phytotherapy but no further medication was prescribed. Furthermore, 57 (53.3%) of 107 switched (outcome 1) to alpha-blockers (*n* = 48), and 5α-reductase inhibitors (*n* = 15); eight were on combination therapy. Twenty (18.7%) patients had clinical progression (outcome 2): acute urinary retention (*n* = 7) and need for surgery (*n* = 13; 10 due to worsened symptoms, eight due to increased residual volume-median 210 mL-; five experienced both); prior to progression, 19 of 20 patients had switched temporarily to drug treatment. Among patients who switched to drug treatment, the median time from presentation to switch was 24 months (3–120). Among those who clinically progressed, the median time from presentation to progression was 54 months (3–72). One case of incidental prostate cancer was found out of 13 patients undergoing transurethral resection of the prostate. Four patients were lost at follow-up.

Age, PSA, prostate volume, and post-void residual volume at presentation were significantly higher in the groups facing treatment switch and clinical progression, as compared to patients on phytotherapy or no therapy at all. These variables did not differ significantly *between* treatment switch and clinical progression groups (Table 1).

### 3.2. Nomograms

In UVA predictive accuracy analyses, age, IPSS, Qmax, prostate volume, PSA, and post-void residual volume were accurate predictors of clinical progression. At MVA, age, IPSS, Qmax, prostate volume, and post-void residual volume were independent predictors of clinical progression (*p* ≤ 0.01). The calibration curves verified the concordance of good-quality models; the areas under the curve (AUCs) of nomograms were 86% and 80% (Figure 1 and Figure 2). In order to avoid a collinearity effect between predictors, prostate volume was not considered in the model (Pearson’s correlation coefficient = 0.631, *p* < 0.001). Two nomograms were modeled to predict treatment switch from phytotherapy (outcome 1) or no therapy to alpha-blockers or 5-ARI and clinical progression (acute urinary retention or need for surgery, outcomes 2) (Figure 3 and Figure 4). The short horizontal lines over predictor scales indicate the 0.1 confidence limits for each score.

## 4. Discussion

LUTS affect 3% of men aged 45–49 years, rising to >30% in men older than 85 years, with a substantial burden on patients and health services [8]. Both BPH symptoms and treatments can potentially have an impact on the quality of life [9]. Phytotherapy is often the first-line treatment of BPH-associated LUTS [10]. The herbal-based approach is perceived as more gentle; many patients aim to avoid synthetic drugs and their related sexual dysfunction, and they appreciate the possibility of self-medication. Furthermore, the implications of the placebo effect are potentially wide and difficult to assess in phytotherapy; its effectiveness might be influenced by a variety of factors other than the specific treatment, such as the consultation process and the degree of empathy of the practitioner with the patient.

However, many decisions regarding a treatment beyond phytotherapy are crucial in the life of patients. Many have one or more comorbidities (i.e., hypertension, diabetes, hypercholesterolemia, etc.) and already take several medications [11]. Decision making in clinics needs to take into account several factors, from recent and past medical history to patient propension to a new drug. In a previous study, we showed that approximately half of the patients with mild LUTS treated with phytotherapy would progress to pharmaceutical or surgical interventions over a 10-year span [12]. We now propose a novel nomogram to help predict the risk of clinical progression patients with BPH-associated LUTS.

Slawin and Kattan proposed a similar nomogram for selecting BPH candidates for dutasteride therapy. They presented an interesting tool incorporating the American Urological Association (AUA) Symptoms and BPH impact indices, prostate volume, PSA, prior alpha-blockers, and dutasteride therapy. However, their nomogram was constructed using phase III pivotal trial data used to establish the safety and efficacy of dutasteride and was designed to identify patients benefitting from dutasteride therapy [13].

Our nomograms are based on a real-world series of patients presenting with mild to moderate BPH-associated LUTS, who were first offered phytotherapy and followed for 10 years. The clinical data incorporated included age, symptoms as assessed by IPSS, prostate volume, PSA, post-void residual volume, and Qmax. All these parameters were significantly associated with progression from phytotherapy to synthetic drugs and were then used to construct the nomograms. As compared to Slawin–Kattan’s nomogram, our nomograms are somewhat simplified. First, only one symptom score (IPSS) is used instead of two; prostate volume was also excluded to avoid a collinearity effect with PSA. Although PSA is considered a less-than-perfect marker of prostate cancer [14], it proved a powerful predictor of clinical progression of BPH. The cumulative incidence of spontaneous AUR increased dramatically with serum PSA above 1.3 mg/mL [15]. Second, age and residual volume were included into our nomogram to offer a wider perspective on the condition of each individual. Although no consensus has been reached [16], worsening residual volume is recognized as a good predictor of AUR in men with LUTS suggestive of BPH [17]. Similarly, residual volume was the most important predictor of treatment switch and of clinical progression in our series. Of note, the median amount of residual volume at presentation was relatively low: 30 mL and 52 mL in patients switching treatment and experiencing clinical progression, respectively. As a consequence, it is suggested that a confirmed residual volume about <50 mL should be followed up closely.

Kozminsky also included residual volume together with age, AUA symptom and BPH impact indices, Qmax, and PSA in his nomogram according to follow-up data at 4.5 years [18]. This study also derived from a randomized trial population, characterized by median older age (62.6 versus 55 years), higher PSA (2.3 versus 1.3 ng/mL), and residual volume (68 versus 10 mL) as compared to ours.

A similar nomogram was proposed by Ganpule et al.; interestingly, the median lobe enlargement was added to IPSS, PSA, peak flow rate, and prostate volume. The need for medical or surgical intervention, globally considered, was the outcome. Age (63 vs. 55 years), prostate volume (36 vs. 32 cc), and PSA (2.0 vs. 1.3 ng/mL) in their series were higher as compared to ours: patients might have been caught in a later phase of their condition, enhancing the need for treatment [19]. De Nunzio created a nomogram able to predict to predict the risk of bladder outlet obstruction; only peak flow and transition zone were considered [20].

In our series, age, PSA, post-void residual volume, and prostate volume were significantly higher in the groups with treatment switch and clinical progression, as compared to patients on phytotherapy or no therapy at all. Conversely, these variables did not differ between the treatment switch and clinical progression groups, likely meaning that it is easier to screen patient candidates for medical treatment for BPH-related LUTS than further screen patients at higher risk for clinical progression.

An important issue is the finding of incidental prostate cancer in patients with BPH-related LUTS. Such patients are generally followed for a long time by the referring urologist. It is crucial that follow-up continues even after clinical progression and surgery. In our study, one case of incidental prostate cancer was found among 13 patients undergoing transurethral resection of the prostate, which is consistent with the rate (6.6%) reported in the literature [21].

We developed a nomogram predicting the switch from phytotherapy or no therapy to alpha-blockers or 5-ARI and a nomogram predicting clinical progression, as previously defined. The cohort which maintained phytotherapy or no therapy had an average probability of switching treatment and progressing clinically of 25% and <5%, respectively, while the cohorts switching therapy or progressing had an average probability of 75% and 40%, respectively. These likelihoods might be tentatively proposed as the thresholds to define patients at high risk (>75% and >40%) or low risk (<25% and <5%) for pharmaceutical or surgical interventions, respectively, on the basis of their clinical features at presentation.

Our study had several limitations. First, a small sample size derived from a single institution was presented. Second, the group of patients switching to synthetic drugs comprised only a minority of patients taking 5-ARI, which might have been useful in preventing clinical progression. However, we propose two nomograms useful to help quantify the risk of frequent events in daily practice in a well-studied cohort of patients followed up for a long interval over their lifetime. Our data await an external validation in order to establish their utility in different clinical settings.

In conclusion, the chance of facing a pharmaceutical or surgical intervention became real after a median interval of 2 and 4.5 years, respectively, in our series of patients with mild to moderate BPH-related LUTS on phytotherapy. The quantification of these risks and the important issues and decisions related to the life and quality of life of patients can be assisted by these novel nomograms, supporting proper counseling and timely therapeutic decisions.

## 5. Conclusions

We developed a novel nomogram to predict the risk of pharmaceutical or surgical interventions for BPH-related LUTS. The patients in this cohort were relatively young men treated with phytotherapy as a first-line approach and followed for 10 years. On the basis of our models, thresholds of >75% and >40% for high risk and <25% and <5% for low risk of pharmaceutical or surgical interventions, respectively, can be proposed.

## Figures and Tables

**Figure 1 ijerph-19-09738-f001:**
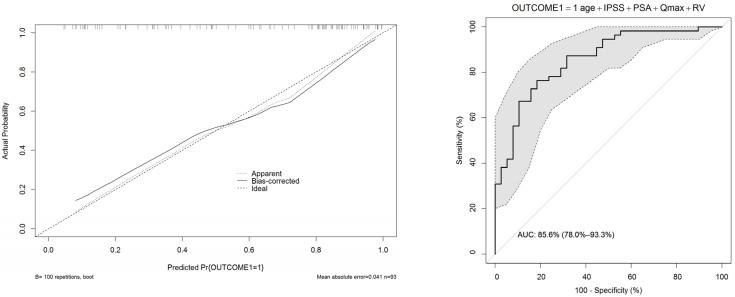
Calibration (**left**) and ROC (**right**) curves of the nomogram predicting the treatment switch from phytotherapy or no therapy to alpha-blockers or 5-ARI for BPH (outcome 1).

**Figure 2 ijerph-19-09738-f002:**
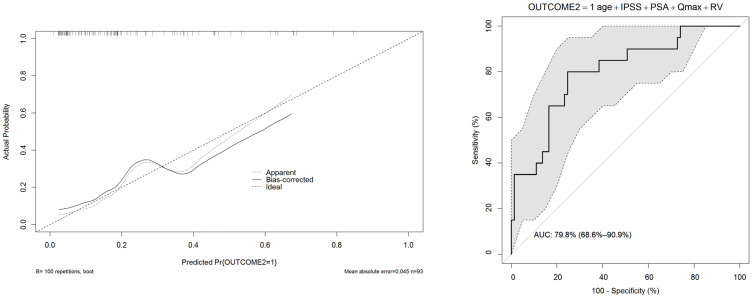
Calibration (**left**) and ROC (**right**) curves of the nomogram predicting clinical progression of BPH (outcome 2 = urinary retention or need for surgery).

**Figure 3 ijerph-19-09738-f003:**
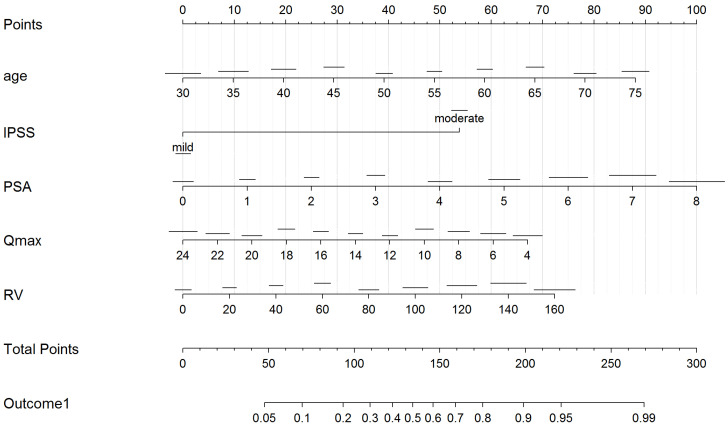
Nomogram predicting the treatment switch from phytotherapy or no therapy to alpha-blockers or 5-ARI for BPH (outcome 1). Legend: IPSS = International Prostate Symptom Score; prostate-specific antigen (PSA); peak flow rate (Qmax); RV = post-void residual volume.

**Figure 4 ijerph-19-09738-f004:**
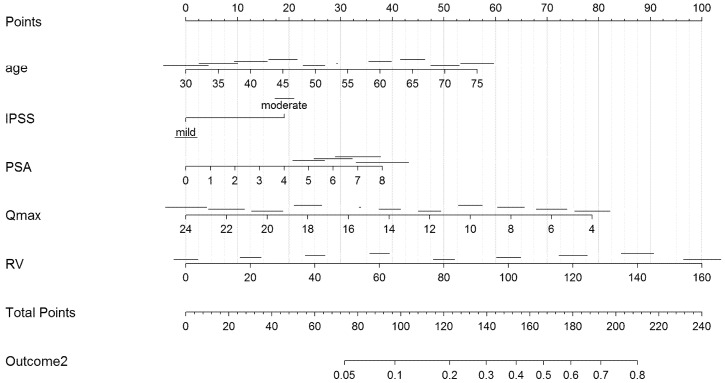
Nomogram predicting clinical progression of BPH (outcome 2 = urinary retention or need for surgery). Legend: IPSS = International Prostate Symptom Score; prostate-specific antigen (PSA); peak flow rate (Qmax); RV = post-void residual volume.

**Table 1 ijerph-19-09738-t001:** Clinical parameters at presentation in 107 patients, divided according to the clinical outcomes (no outcome = no drugs or progression; drug switch = patients switched from phytotherapy to alpha-blockers or 5-alpha reductase inhibitors; progression = urinary retention or need for surgery). The *p*-value refers to the comparison among no outcome, drug switch, and progression groups (Kruskal–Wallis test).

Parameter	Cohort (*n* = 107)	No Outcome (*n* = 50)	Drug Switch (*n* = 37)	Progression (*n* = 20)	*p*
Age (years)	55	49 (9.9)	60 (9.5)	62 (8.0)	<0.001
Prostate volume (cc)	32	28 (12)	40 (17.1)	40 (15.2)	<0.001
Q max (mL/s)	13	16 (3.9)	12 (3.6)	10 (4.8)	0.011
PSA (ng/mL)	1.3	0.85 (1.0)	1.85 (1.5)	1.9 (1.3)	0.001
Residual volume (cc)	10	0 (27)	30 (38.1)	52 (48.9)	<0.001
Mild symptoms (%)	57 (53%)	38 (76%)	13 (35%)	6 (30%)	<0.001

## Data Availability

The data presented in this study are available on request from the corresponding author. The data are not publicly available due to ethical restrictions.

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
