# Peer review of "A Novel Nomogram Based on Initial Features to Predict BPH Progression"

_ijerph, 2022, doi:10.3390/ijerph19159738_

Round 1

Reviewer 1 Report

In the reviewed manuscript, the authors addressed the interesting topic of determining the factors influencing treatment change in patients with BPH. Despite the progress, we do not have adequate indicators of clinical progression that would allow for adequate stratification of patients at the initial stage of treatment, and therefore the attempt to define a nomogram is noteworthy.

The manuscript has potential but requires revisions.

Introduction

 - far too little information about herbal treatment as initiating treatment and being the initial status in the studied group of patients

 - the sentence from the lines suggests that the aim of the study was to determine the nomogram in patients with 10 years of follow-up - probably not so - were all patients followed for 10 years? even those who had an intervention?

 - Lines 50-52 - Is phytotherapy really first line treatment for patients with severe LUTS? Shouldn't be mild to moderate?

Materials and methods:

 - why the minimum period for phytotherapy is 3 months and for pollen extract only 1 month - what is the justification for such a separation?

Results:

Bad presentation of results - this applies to the "Case series" sub-paragraph:

 - Lines - 118-120 - where these numbers came from - there were 107 patients - should be presented in the table

 - Lines - 120-123 - the given data indicate that no patient "a" entered the scenario "c" - is it true? because one can only guess in such a presented version.

 - Line 123 - 20 patients from which group? - that should be precisely defined. This causes differences between the description and the comparison of the data presented in Table 1.

 - 1 patient with antcholinergics - should not be included in the analysis. Everywhere the authors mention 2 groups of drugs, alpha blockers and 5ARI in the results and in the discussion, e.g. lines 215-216

The consequence of this should be that this patient was excluded from the analysis.

Discussion:

 - general remark - the whole paragraph is inconsistent, especially the first part

Lines 174-175 - this sentence should be included in the introduction

Lines 183-184 - the sentence is misleading - again, were all patients followed for 10 years - no one dropped out of observation, died, were patients after surgical intervention also followed for 10 years?

No reference to the placebo effect in the case of phytotherapy - an important aspect especially in relation to the analyzed population where patients did not continue phytotherapy - maybe they did not need it at all.

References - modest

Reviewer 2 Report

The present manuscript entitled “A novel nomogram based on initial features to predict BPH progression” introduces the useful tool for intervention for patients with BPH-related symptoms. The presented data is important as authors have followed patient cohort for 10 years and have presented separate nomograms for different types of outcomes.
The background of the study is sufficiently justified and the results support the conclusions.

The paper in current state requires only minor improvements:

Page 1, lines 24-25
“1/3 of which 25 had clinical progression after a median time of 24 and 54 months, respectively” – it is unclear to what exactly median times of 24 and 54 months refer to.

Page 2, lines 65-70

The part describing BPH and LUTS would be more appropriately placed in Introduction section.

Page 3, line 91

“Symptoms are assessed by patients’ complain and by IPSS” – it is unclear if patients’ complains mentioned are used as part of IPSS or additional to it, as only IPSS is mentioned in nomograms.

Page 3, lines 108-111 / Figures 3,4
There are short horizontal lines present over predictor scales in nomograms, but their meaning is not given in the section describing nomograms.

Page 4, line 134
While previously in text its stated that the significant difference was found only in groups facing treatment switch/clinical progression compared to phytotherapy/no therapy group and not between them, it should be noted in the table to which comparison presented p values refer to.

Page 5,6 – Figures 3 and 4
There is some inconsistence in the use of abbreviations – IPSS is already deciphered previously in the text, PV is deciphered only in nomograms captions, but residual volume is already mentioned several times in the text, and PSA and Qmax are not deciphered anywhere. It is better to either decipher every used abbreviation under nomogram, or in text on first mention.

Page 6, lines 174-176
The part that gives the definition of nomogram would be more appropriately placed in Introduction section.

Also, while authors have presented a solid comparison with previous works using nomograms for progression risk prediction in BPH patients, perhaps it could be expanded – I was able to find a recent work on the subject - Ganpule, Arvind P., et al. "BPH nomogram using IPSS, prostate volume, peak flow rate, PSA and median lobe protrusion for predicting the need for intervention: development and internal validation." American Journal of Clinical and Experimental Urology 9.3 (2021): 202. (https://www.ncbi.nlm.nih.gov/pmc/articles/PMC8303027/). As it describes 2-year follow-up with prediction for intervention of any kind, and contains comparison with several previous works, it could be of useful.

Finally, some minor spelling mistakes need corrections, for example:
Page 1, line 26 – no comma needed after “and post-void residual volume”
Page 2, line 63 – duplicate “were”

Page 2, line 73 – “(UTI)” should be without parentheses

Page 2, line 77 – “undergo” should be in past tense

Page 2, line 84 and Page 3, line 123 – “anticolinergics” should be anticholinergics

Page 3, line 125 – “210ml -;” should be 210ml;

Round 2

Reviewer 1 Report

Authors addressed all comments.